# Tumor Antigens beyond the Human Exome

**DOI:** 10.3390/ijms25094673

**Published:** 2024-04-25

**Authors:** Lisabeth Emilius, Franziska Bremm, Amanda Katharina Binder, Niels Schaft, Jan Dörrie

**Affiliations:** 1Department of Dermatology, Universitätsklinikum Erlangen, Friedrich-Alexander-Universität Erlangen-Nürnberg, 91054 Erlangen, Germany; lisabeth.emilius@fau.de (L.E.); franziska.bremm@uk-erlangen.de (F.B.); amanda.binder@uk-erlangen.de (A.K.B.); jan.doerrie@uk-erlangen.de (J.D.); 2Comprehensive Cancer Center Erlangen European Metropolitan Area of Nuremberg (CCC ER-EMN), 91054 Erlangen, Germany; 3Deutsches Zentrum Immuntherapie (DZI), 91054 Erlangen, Germany; 4Bavarian Cancer Research Center (BZKF), 91054 Erlangen, Germany

**Keywords:** cancer immunotherapy, neo-antigens, neo-epitopes, somatic mutation, alternative splicing, cryptic epitopes

## Abstract

With the advent of immunotherapeutics, a new era in the combat against cancer has begun. Particularly promising are neo-epitope-targeted therapies as the expression of neo-antigens is tumor-specific. In turn, this allows the selective targeting and killing of cancer cells whilst healthy cells remain largely unaffected. So far, many advances have been made in the development of treatment options which are tailored to the individual neo-epitope repertoire. The next big step is the achievement of efficacious “off-the-shelf” immunotherapies. For this, shared neo-epitopes propose an optimal target. Given the tremendous potential, a thorough understanding of the underlying mechanisms which lead to the formation of neo-antigens is of fundamental importance. Here, we review the various processes which result in the formation of neo-epitopes. Broadly, the origin of neo-epitopes can be categorized into three groups: canonical, noncanonical, and viral neo-epitopes. For the canonical neo-antigens that arise in direct consequence of somatic mutations, we summarize past and recent findings. Beyond that, our main focus is put on the discussion of noncanonical and viral neo-epitopes as we believe that targeting those provides an encouraging perspective to shape the future of cancer immunotherapeutics.

## 1. Introduction

Cancer is commonly known as a genetic disease that is caused by DNA mutation [1]. DNA mutations can give rise to so-called neo-antigens. Per definition, neo-antigens are tumor-specific molecules. This means that neo-antigens are not present in healthy cells and are thus foreign to the immune system [2]. Consequently, neo-epitope-specific T cells are not subject to central tolerance, a mechanism in the thymus which ensures the deletion of autoreactive T cells during T-cell development [3]. Two fundamental features of neo-epitopes that are capable of provoking a tumor-targeted cellular immune response are tumor-specificity and MHC-binding potential. In general, the term “neo-epitope” refers to an MHC-binding peptide (MHC: major histocompatibility complex) which is derived from a neo-antigen [4]. It is important to note that the terms “neo-antigen” and “neo-epitope” have different meanings and thus should not be used interchangeably, even if it might be the case in many publications.

It is believed that MHC-presented neo-epitopes help in the prevention of tumor development through immune surveillance mechanisms [5]. For the tumor surveillance system, T cells which secrete interferon gamma (IFNγ) are of particular importance [6]. In line with that, tumors with a high neo-antigen burden have been correlated with increased numbers of tumor-infiltrating T cells (TILs) and prolonged survival of the patients [7,8,9]. This suggests that many TILs are neo-epitope-specific. As neo-epitopes play an important role in the immune defense against tumors, immunotherapies exploiting neo-epitopes are a promising tool in the combat against cancer [10,11]. For example, TILs can be supported in killing neo-epitope-rich tumors via the administration of immune checkpoint inhibitors (ICIs) as TILs have been shown to express high levels of the programmed death receptor-1 (PD-1) [12,13]. Further, neo-epitope-specific TILs can be given to a patient in the form of adoptive cell therapy (ACT). This has been demonstrated to contribute to tumor eradication [14,15]. Additionally, various reviews highlight neo-epitopes as a component of both prophylactic and therapeutic vaccines [4,8,11,16]. All these findings emphasize the significance of neo-antigens in the battle against cancer.

Recently, Nagel et al. suggested that categorization of neo-epitopes into two groups, classic and noncanonical neo-epitopes [10]. Following this discrimination, below we will refer to the classic neo-epitopes as canonical neo-epitopes. This term is used to describe those neo-epitopes which arise as a direct consequence of a DNA mutation, meaning that the mutation is located in an open reading frame (ORF). In most cases, DNA mutations which lead to the formation of canonical neo-epitopes are not shared between patients. Thus, these neo-antigens are considered to be private [17], and respective immunotherapeutics have to be individualized for every patient [3,8]. The only exceptions are reoccurring mutations that give the tumor a selective growth advantage. In this case, the generated neo-epitopes may be shared between multiple patients [18]. This provides an encouraging perspective for the development of “off-the-shelf” drugs which could be readily available for the treatment of different patients with the same neo-epitopes [19]. In comparison to canonical neo-epitopes, noncanonical neo-epitopes are shared between patients more often [17]. Nagel et al. define noncanonical neo-antigens as a combination of all tumor antigens which are inducible and all which are not genetically encoded [10]. Thus, noncanonical neo-antigens arise from aberrant central processes in tumor cells, e.g., aberrant mRNA splicing. Since this is likely to be disease-driving [20], shared neo-epitopes may be generated as a consequence. In this review about canonical and noncanonical neo-epitopes, we would like to set the focus on the latter.

Different cancer identities also share neo-epitopes due to their origin from an integrated viral genome [21,22]. Depending on the process of integration, the viral genome is then transcribed into viral antigens but is also subjected to post-transcriptional modifications of the noncanonical pathways. These shared characteristics of tumor cells can be the basis of immunotherapeutic treatment strategies.

Further, neo-epitopes which are presented on MHC class II molecules have been covered by others [23,24]. To ease understanding of the here-discussed cytotoxic T lymphocyte (CTL) responses, MHC class I molecules are only referred to as MHC or, for the human equivalent, HLA (human leukocyte antigen).

## 2. Canonical Neo-Epitopes

### 2.1. Single Nucleotide Variants

The most studied type of DNA mutation in the context of cancer neo-antigens are undoubtedly single-nucleotide variants (SNVs) within protein-coding regions of the genome [25,26]. Relevant are the so-called missense mutations or nonsynonymous SNVs (nsSNVs). nsSNVs are characterized by a single-nucleotide exchange in a coding region of the DNA, which ultimately results in a single amino acid substitution at the protein level [3,4] (Figure 1).

Although only a single nucleotide and, consequently, only a single amino acid is changed, nsSNVs can cause the formation of neo-epitopes through various mechanisms [27]. These mechanisms act on different levels, including direct as well as indirect ways. On the one hand, an nsSNV can lead to the exchange of an amino acid which influences the T-cell receptor (TCR) recognition directly. In this case, the properties of the peptide–MHC complex are altered due to the substitution of an amino acid residue that faces the TCR binding site [28]. Thus, the complex is recognized by a TCR which is not subject to central tolerance. Further mechanisms, on the other hand, are indirect. A neo-epitope can be generated when a formerly non-MHC-binding peptide gains the ability to form a complex with an MHC molecule. This happens when a nsSNV leads to the exchange of an anchor amino acid which increases the binding affinity of the peptide towards the MHC molecule [25,27,29]. Since the peptide can now be presented on the cell’s surface in the form of a peptide–MHC complex, which is yet again recognized by a TCR that escaped central tolerance, a neo-epitope is formed. Another indirect way is through differential degradation of a peptide in the proteasome. This can be due to an nsSNV if the substituted amino acid causes the preservation or destruction of a proteasomal cleavage site, so that the respective epitope remains uncut instead of being degraded as usual [27]. Again, the peptide can be presented in complex with an MHC molecule and provoke a T-cell-dependent immune response.

Another mechanism by which SNVs can cause the formation of neo-antigens is via the induction of aberrant splicing. According to estimates, up to 15% of all SNVs that are responsible for human genetic diseases impair the splicing reaction [30]. This topic is discussed under Section 3.2. Further, SNVs can influence post-translational modifications and thus result in the generation of neo-epitopes. For example, a pan-cancer analysis based on The Cancer Genome Atlas database (TCGA) found that around 90% of tumors carry SNVs at phosphorylation sites [31]. Post-translational modifications are covered under Section 3.4.

In terms of treatment strategies based on nsSNV-derived neo-epitopes, many advances have been made, and nsSNVs are depicted as the best investigated source of neo-antigens in clinical research [3]. In that regard, melanomas have been studied most extensively as they harbor a high mutational load [32,33]. In cutaneous melanoma, three quarters of all mutations have been reported as SNVs in the form of C > T transitions due to ultraviolet light exposure [34]. Consequently to the high mutational burden, neo-epitope vaccines have been implemented successfully in melanoma trials [35,36,37], which gives hope for the treatment of other highly mutated cancers [33]. Smoking-associated lung cancer, which is characterized by C > A transversions [38], is only one example, and research on respective neo-antigen vaccines is ongoing as reviewed elsewhere [39,40]. The most recent milestone of personalized cancer vaccines which target neo-antigens has been achieved by Moderna and Merck. In February 2023, the U.S. Food and Drug Administration (FDA) granted a Breakthrough Therapy Designation for the mRNA-4157/V940 vaccine in combination with the ICI KEYTRUDA. The treatment is aimed at patients with cutaneous melanoma metastatic to a lymph node, who are at high risk of recurrence. Additionally, one of the key inclusion criteria is the complete resection of the tumor before the start of drug administration. The vaccine consists of the mRNA-4157, a synthetic mRNA which encodes up to 34 neo-epitopes specific for each patient’s individual tumor. Upon injection and endogenous expression of these neo-antigens, the patient’s immune system is primed, and neo-epitope-specific T cells are activated to eradicate the tumor cells. To strengthen this effect, KEYTRUDA is coadministered. KEYTRUDA is the brand name of Merck’s pembrolizumab, a monoclonal antibody which blocks PD-1. This removes the PD-1 immune checkpoint blockade and thus enhances the T-cell-mediated killing of the tumor cells. Within the scope of the phase 2b KEYNOTE-942 trial (ClinicalTrials.gov number: NCT03897881), 157 patients were enrolled. Compared to monotherapy with KEYTRUDA, the adjuvant treatment consisting of the mRNA-4157/V940 together with KEYTRUDA decreased the risk of recurrence or death by 44%. The next steps include the progression to a phase 3 trial and expansion of the vaccine to patients with other tumor types. Importantly, the KEYNOTE-942 trial is the first to demonstrate efficacy of an mRNA-based cancer treatment [41].

The main disadvantage of most nsSNV-targeted immunotherapies is the need for a personalized approach since, according to various reviews, missense mutations are unlikely to cause shared neo-epitopes [25,33,42]. This is the case for so-called passenger mutations, because the cells do not gain a selective growth advantage when they acquire a passenger mutation. Hence, the variability of passenger mutations is high in different tumors and the probability of shared neo-antigens is low. On the contrary, the cells do obtain a selective growth advantage through so-called driver mutations as stated in other review articles [10,43]. Driver mutations are often responsible for the transformation of a healthy cell into a cancer cell and for driving tumorigenesis subsequently. Thus, the likelihood of finding a common driver mutation in different cancer patients is higher than it is for passenger mutations. Consequently, the probability of shared neo-antigens is also higher for driver mutations. Furthermore, driver mutation-caused neo-epitopes depict the better targets for immunotherapy as the cancer cells rely on their expression. This is not given for passenger mutations which is why cancer cells may escape immunotherapy by downregulating respective neo-antigens [10]. The most prominent example for driver mutations are *RAS* mutations, e.g., *KRAS* and *NRAS* [44]. According to statistical analyses, approximately 14% of all cancer patients harbor a mutant *KRAS*, making *KRAS* the most frequently mutated oncogene amongst all cancers [45,46]. In a majority of the cases, *KRAS* mutations result in the KRASG12D mutant [45]. It describes a single amino acid substitution of glycine (G) with aspartic acid (D) at the twelfth position of the KRAS oncoprotein [47]. In the context of HLA-C*08:02 KRASG12D, neo-epitopes have been reported as “high-quality” neo-epitopes since they have high HLA-C*08:02 binding affinity while peptides derived from wild-type *KRAS* have low affinity for the same. This minimizes the risk of off-target effects in ACTs [48]. Within the scope of ACTs, HLA-C*08:02-restricted TCRs with KRASG12D-specificity have been used successfully. In one patient, this treatment led to the objective regression of metastatic pancreatic cancer [49]. Objective regression was also seen for all seven lung metastases of a patient with metastatic colorectal cancer [47]. Both reports are part of an ongoing phase 2 clinical trial of neo-epitope-directed ACT in metastatic solid cancers (ClinicalTrials.gov number: NCT01174121). In the case of colorectal cancer, a population-based study found 2.3% of all recruited patients (3734 HLA-typed patients) to be KRASG12D- and HLA-C*08:02-positive [50]. This underlines that KRASG12D is highly relevant in the context of shared neo-antigens. Also, attempts at broadening the cohort of treatment-eligible patients are promising. On the one side, this includes a combinatorial approach, which connects phylogenetic and structural analyses to decipher KRASG12D neo-epitopes with other HLA-C-specificities [51]. On the other side, this includes TCR engineering for an HLA-A*11:01-restricted KRASG12D neo-epitope [45]. In addition, research on cancer vaccines in the setting of *KRAS* mutant cancers is ongoing as well. Advances in terms of the latter are reviewed elsewhere [52]. Next to *KRAS* mutations, other driver nsSNVs have been reported for *BRAF*, *EGFR*, *MYD88*, and *TP53*, among others. These and their role in the generation of shared neo-antigens have also been reviewed elsewhere [53].

### 2.2. Insertions and Deletions

Another commonly investigated source of neo-antigens are insertion and deletion mutations of the DNA, collectively called indels. Insertion and deletion mutations describe the addition and removal of one or more nucleotides, respectively. If multiples of three bases are inserted or deleted, the indel is considered to be in-frame. In-frame indels can cause the formation of a neo-epitope through two ways: (1) the insertion of multiple random nucleotides leads to multiple new amino acids at the protein level and, thus, to neo-epitopes and (2) the deletion of multiple nucleotides causes a de novo fusion of two DNA sequences which leads to a new amino acid sequence at the protein level and thus to neo-epitopes. The opposite to in-frame indels are out-of-frame indels. These entail insertions and deletions which are not multiples of three nucleotides. Hence, out-of-frame indels result in a frameshift of the reading frame. Consequently, the subsequent codons and thus the amino acid sequence are changed completely until running into a stop codon [54]. Due to the drastic alteration of the protein, many neo-epitopes may be generated. However, it should be mentioned that frameshift indels are often transcribed into mRNAs which are degraded prior to translation. This is the case if the mRNA harbors a premature stop codon since such an mRNA is subject to nonsense-mediated decay (NMD). Generally, NMD functions to degrade RNAs with a premature stop codon during the pioneer round of translation [10,55]. So, all in all, indels can lead to the formation of neo-antigens if the encoded RNAs do not contain premature termination codons. Alternatively, indel-derived transcripts may also generate neo-antigens in the case of NMD escape. Exemplary possibilities of NMD escape include stop codon readthrough and alternative initiation of translation [56] (Figure 1).

In general, indels do not occur frequently and are observed significantly less often than SNVs. The highest burden of indels can be seen in renal cancers, peaking for the clear cell renal cell carcinoma (ccRCC). But even ccRCC only shows a fraction of 12% frameshift and 3% in-frame indels compared to the indel and SNV count combined [26]. Frameshift indel-derived neo-antigens have been reported to have multiple advantages over nsSNVs. First, the frameshift indel-caused proportion of high-affinity neo-epitopes is predicted to be three-times higher. Furthermore, only one out of nine neo-epitopes is predicted to be recognized by a TCR with a high affinity towards self-peptides. Consequently, eight out of nine neo-epitopes are recognized by T cells which are not subject to central tolerance [57]. In addition, the ability of frameshift indel-derived neo-epitopes to bind to MHC molecules was experimentally shown to be significantly increased [26]. All in all, most studies have focused on neo-antigens derived from frameshift indels and only little work has been realized on neo-antigens caused by in-frame indels. Just recently, prediction analyses [58] suggested that in-frame indels contribute to the quantity of neo-epitopes to nearly the same extent as out-of-frame indels. These analyses were based on the Jurkat leukemia cell line and the MHC-binding capability of neo-epitopes caused by SNV, frameshift indel, in-frame indel, and gene fusion was calculated. Within this group of neo-epitopes, 3.8% and 3.3% were predicted to be frameshift- and in-frame indel-derived, respectively [58]. It should be mentioned that a study on ccRCC failed to demonstrate an experimental T-cell response against in-frame indel-caused neo-epitopes, whilst the same was present for both SNV- and frameshift indel neo-epitopes. However, the absence of immunogenic in-frame indel neo-epitopes could have been due to the small cohort size of six patients only [26].

Like in the case of nsSNVs, most indel-targeted immunotherapies require a personalized approach because the mutation-derived neo-epitopes are patient-specific [26,57]. The only exception in which frameshift indels cause the same neo-antigens in different patients is a deficiency in the mismatch repair (MMR) system [55,59]. As the MMR mechanism is important to correct mistakes which are made during DNA replication, an MMR defect leads to the accumulation of mutations. Most often, these mutations occur in regions which are particularly prone to errors, e.g., microsatellites. Microsatellites are stretches of one to six base pairs which are repeated 10- to 60-times and located throughout the human genome. Due to their repetitive nature, these regions are at high risk for DNA polymerase slippage which causes frequent replication mistakes [60]. In healthy individuals, the resulting mutations are generally repaired effectively. In MMR-deficient patients, however, microsatellite-instable (MSI) mutations accumulate leading to a heightened risk for the development of tumors. MSI-high (MSI-H) associated cancers include, but are not restricted to, colorectal, endometrial, and stomach cancers. Promising is that MSI-originated neo-epitopes which are both immunogenic and shared between patients have been identified. Thus, MSI-H related tumors are suitable candidates for the development of “off-the-shelf” neo-epitope vaccines [55,59]. Studies have shown the potential of such vaccines both in a prophylactic and therapeutic application. A prophylactic vaccine successfully prevented the development of Lynch Syndrome, a hereditary form of colorectal cancer, in a mouse model [61], and a clinical trial (ClinicalTrials.gov number: NCT01461148) demonstrated immunologic efficacy of a neo-epitope vaccine in patients with a history of late-stage colorectal cancer diagnosis [62]. On the therapeutic side, immunologic efficacy was shown in a clinical trial of metastatic colorectal cancer (ClinicalTrials.gov number: NCT03152565) but failed to translate to tumor eradication in 89% of patients (17 out of 19) [63]. Hence, the successful prevention and treatment of MSI-H tumors with vaccines targeting frameshift-derived neo-epitopes is a not yet fulfilled task.

### 2.3. Gene Fusions

The molecular properties of gene fusions have been studied for around 40 years in the context of cancer [64], but a standardized nomenclature has only recently been suggested by the HGNC (HUGO Gene Nomenclature Committee). The introduced system uses a double colon to connect fused genes and thus symbolizes their formation through chromosomal breakage (first single colon), which is followed by the fusion itself (second single colon) [65]. Since the gene fusion can cause the formation of new open reading frames (ORFs), neo-antigens may arise if the encoding mRNA is not subject to NMD [18]. Compared to in-frame fusions, frameshift fusions have the potential to give rise to more immunogenic neo-epitopes [66,67]. The underlying mechanisms are categorized into balanced and unbalanced chromosome rearrangements. Further, it is differentiated between translocation, insertion, inversion, and deletion. These mechanisms are discussed in detail elsewhere [67,68,69] (Figure 1).

Compared to SNVs and indels, gene fusions occur less frequently in cancer. For example, renal and MSI-H cancers have the highest indel burden among all types of cancers but the lowest burden of gene fusions [67]. However, even though the number of mutations caused by gene fusions is low, fusion-derived neo-epitopes have been predicted to have higher immunogenic capacities than SNV- and indel-derived neo-epitopes. This was shown in a pan-cancer analysis based on the TCGA database [18]. According to these prediction analyses, gene fusions give rise to eleven-times more potential neo-antigens than SNVs and indels. On the downside, only 5.8% of the potential neo-antigens were shared between patients of the TCGA cohort, and, on top of that, shared neo-epitopes were of low immunogenicity. This makes the design of respective “off-the-shelf” immunotherapies unfavorable, i.e., drugs would have to be personalized [18].

In the context of gene fusion, commonly studied cancers are chronic myeloid leukemia (CML) and synovial sarcoma since both are characterized by a fusion construct. In the case of CML, it is a BCR::ABL fusion protein, and in the case of synovial sarcoma, it is SYT::SSX1 [70,71]. In both cases, in vitro studies demonstrated the successful induction of CTL responses against fusion-derived peptides [72,73]. However, clinical translation has not been successful yet [74,75].

The treatment options related to the above-described mechanisms of epitope formation are summarized in Table 1.

## 3. Noncanonical Neo-Epitopes

### 3.1. Expression of Noncoding DNA Regions

Noncoding DNA regions have been shown to be highly relevant for the generation of tumor-specific antigens. In fact, a study utilizing a proteogenomic approach found 90% of tumor-specific antigens to have their origin in noncoding DNA regions. Within that scope, the main source of tumor-specific antigens were endogenous retroelements (EREs) [17]. EREs belong to the family of transposable elements (TEs), and their dysregulation is a fundamental characteristic of many tumors [76]. EREs can be subdivided into long terminal repeat elements (LTRs) and non-LTRs. The former are composed of human endogenous retroviruses (hERVs), while the latter consist of LINEs (long interspersed elements), SINEs (short interspersed elements), SVA retrotransposons, and processed pseudogenes [77]. A publication studying hERVs [78] showed clustering of RNA sequencing data between various cancers. Since similar RNA profiles are expected to result in a similar protein expression pattern, hERVs might harbor the potential to generate shared neo-antigens. Focusing on ccRCC, the same study found epitopes with MHC-binding capability and confirmed their immunologic role in tumor biopsy samples where they detected epitope-specific T cells. Tumor specificity was indicated via comparison with T-cell populations in the peripheral blood of healthy donors [78]. Further hints at TE-derived neo-epitopes are demonstrated in a pan-cancer analysis of the TCGA database [79]. In addition to hERVs, the study also shows LINEs, SINEs, and SVA loci to encode peptides with the potential to form complexes with MHC [79] (Figure 2D).

The before-mentioned group of processed pseudogenes describes a subgroup of pseudogenes. Pseudogenes are degenerated genes that have lost their primary purpose over the course of evolution. Processed pseudogenes are pseudogenes that have an RNA intermediate while unprocessed pseudogenes do not. According to a review, it is estimated that around 10% of the genome’s pseudogenes are transcribed [80]. Translation of pseudogenes into immunogenic proteins has been reported in the context of nonsmall-cell lung cancer (NSCLC). Also, tumor-specificity was given as proven by the absence of these proteins in healthy controls. However, the corresponding study only investigated the humoral immune response and thus did not test for MHC-binding ability [81]. In general, it is believed that pseudogenes mainly serve a regulatory function, e.g., in the form of long noncoding RNAs (lncRNAs). Of all known human lncRNAs, approximately 19% are pseudogene-derived [80].

A prominent example of a lncRNA in the context of cancer biology is the *Plasmacytoma Variant Translocation 1 (PVT1)*. *PVT1* encodes a lncRNA and is located in proximity to the *myelocytomatosis* (*MYC*) oncogene [82]. The *MYC* oncogene is known to be overexpressed in many cancers and to be involved in tumorigenesis. Further, high expression levels of *MYC* have been associated with particularly aggressive cancers, which are summarized in a review [83]. It has been shown that *PVT1* expression is crucial for high MYC levels in two human breast cancer cell lines. Also, an analysis of the TCGA database confirmed this observation as over 97% of *MYC*-upregulated cancers had coincreased *PVT1* copy numbers [82]. A study investigating six MMR-proficient colorectal cancer samples detected 7.5-times higher levels of the lncRNA *PVT1* than in the four matched normal colorectal mucosa tissue samples. Further, they found *PVT1* to be translated into a peptide capable of binding to MHC. On top of that, PVT1-specific CTLs were present both in the form of TILs and circulating T cells in the cancer patients. Even though true tumor-specific expression of *PVT1* was not given, the PVT1 protein was 15.9-fold enriched in cancer samples compared to the controls [84]. This indicates that the PVT1 antigen is a potential target for immunotherapy, but more research is required to prevent autoimmune side effects. In addition, future work has to determine if this tumor-dominant expression of *PVT1* is a common phenomenon of *MYC*-copy-increased cancers as that might pave the way for the development of “off-the-shelf” vaccines against many cancers.

### 3.2. Splicing

Splicing is a central mechanism in protein expression which requires tight regulation. It is catalyzed by large ribonucleoprotein complexes (RNPs), so-called spliceosomes. Spliceosomes consist of five small nuclear RNAs (snRNAs: U1, U2, U4, U5, and U6) and over 100 core proteins. To splice a pre-mRNA into a mature mRNA, three sites of the introns are of particular importance: the 5′ splice site (5′ss), the 3′ splice site (3′ss), and the branch point sequence (BPS). The spliceosome is assembled upon recognition of these sites. Alternative splicing occurs with a dependency upon which splice sites are bound by a single spliceosome. Which splice sites are used can be controlled by *cis*-regulatory elements since they recruit *trans*-acting factors that facilitate or block the use of a splice site. Thus, *cis*-regulatory elements can act as splicing enhancers or silencers. The resulting possibilities of alternative splicing include alternative 5′ss usage, alternative 3′ss usage, exon skipping, and intron retention [85,86,87]. Regarding the generation of neo-antigens through aberrant splicing, two possibilities have been reported most. On the one hand, a mutation which destroys or creates a splice site (ss) may lead to differential alternative splicing which changes the protein expression pattern. These mutations are summarized as *cis*-acting mutations (see Section 3.2.1). On the other hand, *trans*-acting mutations and defects of the splice machinery describe mutations of the proteins involved, e.g., a mutated splice factor, and can also cause the formation of neo-antigens (see Section 3.2.2) [71,88]. Hereafter, defects of the splicing machinery are meant to be included in the term of *trans*-acting mutations. The major difference between *cis*- and *trans*-acting mutations is the probability of shared neo-antigens. While a splice site mutation is likely private, the chance of shared neo-antigens due to a mutation in a splice factor is much higher [25,89] (Figure 2C).

In addition to *cis*- and *trans*-acting mutations, aberrant splicing may also be due to post-translational modifications or epigenetic regulation of splice factors. This topic has been reviewed elsewhere [90].

Alternative splicing is not only a central mechanism in protein expression but has also been found to occur approximately 1.2-times more often in tumor samples than in matched healthy samples [91]. Furthermore, splicing alterations are considered to be a central hallmark of cancer because of their oncogenic potential [92,93]. A publication by Oka et al. [94] validated the relevance of aberrant splicing in the generation of neo-antigens experimentally. Their study focused on NSCLC and the experimental approach comprised three levels: (1) NSCLC cell lines, (2) clinical lung cancer specimens, and (3) mice in vivo immunizations. For the cell lines (1), they determined aberrantly spliced mRNAs through full-length cDNA sequencing. Since this method comes with a high error rate, splice junctions were confirmed with short-read RNA sequencing. To confirm translation of aberrant splicing isoforms, proteome analyses were performed. Interestingly, they detected protein expression of a transcript which had previously been labelled as not having an ORF. This transcript was derived from alternative 5′-splicing of the *KRT7* gene and was found in four of the 22 analyzed NSCLC cell lines. According to prediction analyses, this splicing isoform can give rise to multiple peptides with MHC-binding capability. These candidate MHC-binders are described as neo-epitopes in spite of lacking demonstration of their tumor-specificity. For the clinical samples (2), the detection of aberrant splicing isoforms was accomplished via cDNA and RNA sequencing again. Here, tumor-specificity was regarded, and the applying inclusion criterion was to have at least two-times more isoforms in the tumor samples compared to nontumor samples. Again, MHC-binding ability of splicing isoform-derived peptides was predicted to identify neo-epitope candidates. Of these, 17 peptides restricted to a certain MHC molecule were selected to immunize transgenic mice (3). The immunogenic potential of these peptides was demonstrated, as neo-epitope-specific T cells were detected for eight peptides in a follow-up assay. This provides experimental evidence of immunogenic neo-epitopes which are generated by aberrant splicing isoforms [94]. Still, future research should keep in mind that the neo-epitopes presented here were not fully tumor-specific.

*Cis*- and *trans*-acting mutations are the two most reported ways of splicing-dependent neo-epitope generation. As stated, intron retention is one possible way for the formation of neo-epitopes. Including neo-epitopes resulting from intron retention into the pool of neo-epitopes caused by somatic mutation has been predicted to increase the quantity of neo-epitopes by a factor of 1.7 [95]. In rare cases, even “normal” intron retention, i.e., intron retention that happens under physiologic cell conditions, can cause the expression of neo-antigens. This was the case for a melanoma patient where the retained intron harbored an nsSNV that was tumor-specific [96]. However, intron retention may not always lead to tumor-specific antigens. For example, intron retention led to similar levels of antigen in a melanoma cell line compared to melanocytes in another study [97]. Hence, in the latter case, intron retention did not generate a neo-antigen. This shows the importance of differentiating between neo-antigens depending on tumor-specific splicing and splicing-derived antigens which are not tumor-specific and thus, per definition, not neo-antigens. It is clear that neo-antigens can be generated if aberrant splicing occurs as a direct result of somatic mutation, i.e., a splice site mutation or a defect in the splicing machinery [71,88].

#### 3.2.1. Cis-Acting Mutations

Mutations at splice sites are often overlooked and falsely classified as missense or silent mutations making them an underestimated source of neo-antigens [85,88,98]. In fact, up to 96.8% of SNVs which are located in proximity to the 5′ss have been predicted to cause aberrant splicing as stated in a review article [99]. The 5′ss is a conserved sequence complementary to the 5′ terminus of the snRNA U1. A mutation affecting the 5′ss can render U1 unable to recognize the ss. This can lead to whole exon skipping if another authentic 5′ss is bound by the spliceosome instead [100]. The 3′ss is a consensus sequence bound by the U2 auxiliary factor (U2AF) which initiates complex formation with the snRNA U2. Together with other factors, this makes up a small nuclear ribonucleoprotein complex called U2 snRNP [101]. Analogous to a 5′ss mutation causing whole exon skipping, if U1 binds to a different authentic 5′ss, a 3′ss mutation can cause whole exon skipping if U2 uses a different authentic 3′ss [102]. However, other consequences of a mutation at a 5′ss or 3′ss exist as well. Alternative to the usage of a different authentic ss, a so-called cryptic ss may be used. Per definition, a cryptic ss matches the consensus sequence of the authentic ss but is only chosen for splicing if the pre-mRNA harbors a mutation, e.g., in the form of a mutated authentic ss. The cryptic ss can either be located in the intron or the exon. If a cryptic ss is used instead of the mutated authentic ss the splicing reaction results in partial intron retention or partial exon skipping, respectively [99,100,103]. Also, complete intron retention is possible in the case of a small intron. In this scenario, no splicing occurs at the mutated ss whatsoever, causing the named intron inclusion [104]. On the contrary to mutations destroying ss, mutations in an intron or exon may create ss. However, a de novo created ss is only used for splicing in combination with a cryptic ss but not an authentic ss. Depending on the positions of the ss, different alternative mRNAs can be formed. De novo created ss which are not used for splicing are named pseudo ss [100]. Aside from the 5′ss and 3′ss, a third site is relevant for splicing, the branch point sequence (BPS). Mutations at the BPS can lead to whole or partial intron retention or exon skipping. According to a review, only few mutations at the BPS have been discovered [102].

The mis-annotation of *cis*-acting mutations was emphasized in a study which investigated ss-creating mutations in the TCGA database. According to their analyses, 26% and 11% of the previously classified missense and silent mutations account for ss-creating mutations, respectively [98]. They also found that a single splicing isoform derived from a de novo ss can give rise to over 40 neo-epitopes which, based on their MHC-binding potential, are more immunogenic than neo-epitopes formed as a consequence of non-synonymous mutations. Examples of such alternative splicing isoforms include *SMARC1*, *KDM6A*, and *NOTCH1* [98].

#### 3.2.2. Trans-Acting Mutations

Here, *trans*-acting mutations encompass mutations in *trans*-acting splice factors and mutations in factors of the spliceosome. Mutations in the splicing machinery can influence ss selection as well as splicing efficiency and lead to the expression of neo-antigens. Initially, mutations in components of the spliceosome were discovered in hematological malignancies as reported in a review [88]. Within this group of cancers, noncanonical neo-antigens are of notable interest since the expression of canonical neo-antigens is rather limited. Thus, noncanonical neo-epitopes may expand potential immunotherapy targets drastically. This is particularly the case for acute myeloid leukemia (AML), acute lymphocytic leukemia (ALL), and chronic lymphocytic leukemia (CLL), all of which harbor a low SNV burden and, in multiple reviews, are described to generate canonical neo-antigens only occasionally [33,42]. However, aberrant splicing-derived neo-antigens may also broaden the treatment options in solid tumors where *trans*-acting mutations have been found as well [88]. A pan-cancer analysis of the TCGA database [91] has demonstrated this for breast cancer (BRCA) and ovarian serous cystadenocarcinoma (OV) patients. On average for both cancer types, only 30% of the patient samples had at least one SNV-derived neo-epitope while 75% of the patient samples had at least one splicing-derived neo-epitope. In addition, they discovered putative shared neo-epitopes caused by alternative splicing. Out of 63 samples in total, ten neo-epitopes were recurring within either BRCA or OV patients and, on top of that, five neo-epitopes were shared between the two cancer cohorts [91]. This highlights the relevance of noncanonical neo-antigens in cancer research as shared neo-epitopes give hope for the development of “off-the-shelf” drugs.

The splicing factor which has been found to be most commonly mutated in different cancers is the Splicing Factor 3b Subunit 1 (SF3B1). Mutated variants of SF3B1 (SF3B1^MUT^) are seen in hematological malignancies like CLL and myelodysplastic syndromes (MDS) but also in solid tumors such as uveal melanoma (UM), mucosal melanoma, BRCA, pancreatic, and prostate cancer [105,106]. Among these, SF3B1^MUT^ is most prevalent in CLL, MDS, and UM [107]. The most prominent example of SF3B1^MUT^ is an amino acid substitution of lysine (K) with glutamic acid (E) at the position 700, which yields the SF3B1^K700E^ variant [89,105,108,109]. SF3B1^K700E^ is shared both within and between some CLL [109], MDS, BRCA [110], and pancreatic cancer [109] patients. Aside from K700, other commonly mutated sites of SF3B1 include K666 and R625 (arginine in position 625), with R625 being the most frequently mutated amino acid of SF3B1 in UM [89,107]. In terms of its function, SF3B1 fulfils a central role in the splicing reaction. As a core component of the snRNP U2, SF3B1 binds to the BPS of the pre-mRNA and is essential for 3′ss recognition [89,111]. In most cases of SF3B1^MUT^, an alternative BPS is bound, and the authentic 3′ss can no longer be recognized due to steric hindrances. In turn, this favors the usage of cryptic, intronic 3′ss [106,110,112]. Since many pre-mRNAs depend on U2-mediated splicing, a mutation of SF3B1 can induce significant changes at the transcriptomic and proteomic level. According to computational analyses, the cryptic 3′ss which are used most often by SF3B1^MUT^ are positioned in a nontriplet distance of nucleotides from the authentic 3′ss. Thus, the likelihood of the splicing reaction resulting in frameshift transcripts is high. This can lead to NMD of the mature mRNA if a premature termination codon is present [110]. On the contrary, SF3B1^MUT^ can also result in more efficient splicing compared to its wild type (SF3B1^WT^) as seen in a study on UM [108]. Generally, both SF3B1^MUT^ splicing-derived transcripts which are not subject to NMD and transcripts that escape NMD harbor the potential to form neo-antigens. Support for NMD escape of SF3B1^MUT^-generated transcripts was presented in a study by Schischlik et al. [113]. In a patient cohort with myeloproliferative neoplasms, a group of hematological cancers, they analyzed RNA sequencing data of *SF3B1^MUT^*-patients to identify potential neo-epitopes. For that, they selected 16 frameshift transcripts with a premature termination codon and three in-frame transcripts. All 19 transcripts were predicted to cause major changes in the encoded amino acid sequences. Astonishingly, only one of the 16 frameshift transcripts seemed to be NMD-sensitive. Further, one such transcript was significantly upregulated in spite of its premature termination codon. In an in vitro assay, 13 of the 19 peptides (68.4%) were confirmed to have MHC-binding capability. However, the corresponding splice junctions could also be found in *SF3B1^WT^*-patients, albeit at lower expression rates compared to *SF3B1^MUT^*-patients. Thus, they might still be of relevance as potential immunotherapy targets despite lacking true tumor-specificity [113].

Off track from *trans*-acting mutations having the potential to generate neo-antigens, pharmacologic blockage of splicing with a spliceosome inhibitor may also trigger the production of neo-antigens [114]. In particular, this has been shown in a study by Lu et al. [115] which focuses on Indisulam. Indisulam is an inhibitor that targets the splicing factor RBM39 (RNA binding motif protein 39) for ubiquitin-mediated degradation, thereby driving aberrant splicing [116,117]. If Indisulam is administered to murine cancer cell lines at low doses, RBM39 levels decrease but cell growth remains unaffected. Upon engraftment into immunocompetent mice, however, blockage of tumor growth is achieved. Further in vitro and in vivo experiments suggested a mechanism depending on Indisulam-induced neo-epitope generation and subsequent tumor killing by neo-epitope-specific CTLs. This indicates that Indisulam can affect splicing in a manner that leads to the formation of immunologically meaningful neo-epitopes. In addition, Lu et al. provide evidence for the low toxicity of Indisulam administration which confirms the potential for clinical translation [115]. Conversely, the study demonstrates aberrant splicing as a possible source of immunogenic neo-epitopes.

For a comprehensive overview, the topic of aberrant splicing may be subdivided into *cis*- and *trans*-acting mutations. *Cis*-acting mutations can either destroy or create splice sites and thus cause (partial) intron retention or (partial) exon skipping. *Trans*-acting mutations affect the components which drive the splicing reaction and thus influence splice site selection as well. Regarding the implication of aberrant splicing in the context of cancer neo-epitopes, in vitro and in vivo experiments have proven the contained potential to generate immunogenic neo-epitopes. In particular, *trans*-acting mutations provide an encouraging perspective for the development of “off-the-shelf” immunotherapeutics which could be used for the treatment of various cancers.

### 3.3. Translation

Translation is a central mechanism in cells that is catalyzed by large complexes, the ribosomes. Given the fundamental role of translation, it comes as no surprise that multiple reviews name aberrant mRNA translation as a common characteristic of many cancers [118,119]. Aberrant mRNA translation can mainly be divided into two categories: (1) translation of putatively noncoding sequences and (2) translation of coding sequences in a noncanonical fashion [120]. The first includes the translation of noncoding DNA regions (see Section 3.1) as well as the translation of aberrantly spliced mRNAs, e.g., mature mRNAs which still contain intronic regions (see Section 3.2). The second can affect all three phases of translation, i.e., the initiation, elongation, and termination (Figure 2B).

#### 3.3.1. Initiation of mRNA Translation

Canonical mRNA translation is initiated at the cognate start codon which is composed of the nucleotide triplet adenine, uracil, and guanine (AUG) [121]. Noncanonical initiation of translation can happen through various mechanisms [120]. One possibility is via a scanthrough of the first AUG and initiation of translation at the second AUG which leads to the translation of an alternative ORF (aORF). An aORF can only cause the formation of a neo-antigen if it is shifted towards the original ORF, i.e., if the protein sequence is altered. Additionally, a frameshifted aORF may not contain a premature stop codon or, otherwise, it would be subject to NMD (see Section 2.2). Another way of noncanonical initiation of mRNA translation is the usage of non-AUG start codons, particularly of near-cognate start codons, i.e., triplets that differ from AUG by a single base. Translation beginning at near-cognate start codons can cause the translation of frameshifted aORFs as well. Also, near-cognate start codons may be partially or completely located outside the original ORF of the mRNA and thus the aORF may contain 3′ or 5′ untranslated regions (UTRs) [122,123]. An extensive study on the translatome of mouse embryonic stem cells found that most non-cognate start codons initiate translation of aORFs containing 5′ UTRs [124]. Knockout experiments have demonstrated that the absence of a certain ribosomal protein (RP) increases the use of non-AUG start codons [125]. This suggests mutated RPs as a potential reason for noncanonical initiation of mRNA translation. Indeed, mutated RPs have been identified by multiple reviews in various cancers and are proposed to play a role in cancer development [122,126]. Importantly, noncanonical initiation of mRNA translation is known as a source of peptides with MHC-binding capability, which is reviewed in more detail elsewhere [127]. In essence, scanthrough of the first AUG has been found to generate aORFs which encode CTL epitopes on the one hand [128,129]. On the other hand, initiation of mRNA translation at a near-cognate start codon can cause the same. For example, an analysis of all MHC-bound peptides of two RCC samples revealed a peptide which was presumably derived from noncanonical initiation of translation at the near-cognate start codon CUG. The sequence of the peptide was located in the 5′ UTR of the *vascular endothelial growth factor* (*VEGF*) in an in-frame distance from the canonical ORF. Further, the peptide was identified as a T-cell epitope and potent immunogenicity was demonstrated in in vitro experiments. However, the same epitope was also detected in healthy kidney tissue, albeit at lower levels. The ratio of tumor to control burden was 5.95-fold in average for the two RCC samples [130]. Thus, tumor-specificity was not given but the tumor-dominant expression still gives hope for future research in this field.

#### 3.3.2. Elongation of mRNA Translation

During the elongation reaction, aberrant mRNA translation can be caused via two different mechanisms: (1) ribosomal frameshifting and (2) codon reassignment. Ribosomal frameshifting (1) describes the slippage of the ribosome during the process of translation. The ribosome can either slip one nucleotide towards the 5′ end (−1) or one nucleotide towards the 3′ end (+1). Consequently, ribosomal frameshifting results in the translation of aORFs which are out-of-frame compared to the original ORF [131]. Ribosomal frameshifting has mostly been studied within the scope of programmed ribosomal frameshifting (PRF), which is commonly exploited by retroviruses for protein production [132]. However, the research group of Reuven Agami has found that cancer cells utilize ribosomal frameshifting in a similar fashion as viruses. In cancer cells, ribosomal frameshifting is induced to maintain the expression of proteins when essential amino acids are lacking. This form of PRF was termed sloppiness [133]. Various studies under the correspondence of Agami [133,134,135] have focused on the essential amino acid tryptophan, because tryptophan shortage is known to play a role in tumor progression. Both the lack of tryptophan and the tumor progression itself can be consequences of an immune escape mechanism mediated by the expression of IDO1 (indoleamine-2,3-dioxygenase 1). Many tumors induce the expression of IDO1 upon IFNγ stimulation, which is a cytokine commonly secreted by TILs. IDO1 is an enzyme that catabolizes tryptophan and thus its expression leads to the decrease in tryptophan levels. Furthermore, multiple reviews state that the product of the reaction, kynurenine, can suppress T cells directly [131,136]. Given this background, exposition of a melanoma cell line to IFNγ was found to cause IDO1-dependent depletion of tryptophan and sloppy mRNA translation. Furthermore, 81 sloppiness-resulting peptides were identified and the capability to activate CTLs was demonstrated for two peptides. Regarding tumor-specificity, no experiments were performed in the study [134]. In the context of tryptophan shortage, sloppiness was also detected in many other human cancer cell lines including skin, breast, ovarian, lung, and colorectal cancer, while no sloppiness was seen in any of the tested noncancerous cell lines [133]. The latter hints at tumor-specific induction of sloppiness which, in relation to the before-mentioned study [134], is promising for the development of neo-epitope-targeted immunotherapies. Furthermore, the comparison of sloppy and nonsloppy cancer cell lines revealed a putative association of sloppiness with certain oncogenic mutations. Noteworthy, cancer cell lines with a high degree of sloppiness were positive for prominent oncogenic mutations in *EGFR* (*epidermal growth factor*), *RAS*, and *RAF* [133]. If sloppy translation is associated with specific mutations, hope exists that, in different patients with the same mutation, translation is altered in the same way. Conversely, this could lead to shared neo-antigens.

In addition to ribosomal frameshifting, aberrant translation elongation can also occur through codon reassignment (2). In the same setting of IFNγ-induced, IDO1-mediated tryptophan depletion, a thorough study [135] demonstrated translation of both out-of-frame and in-frame proteins. Evidence suggests that these proteins were derived from ribosomal frameshifting and codon reassignment, respectively. Instead of tryptophan, the in-frame-generated proteins carried phenylalanine. Peptides containing phenylalanine (F) in place of tryptophan (W) were termed W>F substitutants. Analysis of the Clinical Proteomic Tumor Analysis Consortium (CPTAC) database revealed enrichment of W>F substitutants in various cancers, e.g., hepatocellular carcinoma, squamous-cell lung cancer, as well as head and neck squamous-cell carcinoma. Moreover, W>F substitutants were shown to induce a potent T-cell immune response in vitro, but the expression of W>F substitutants was only tumor-dominant in comparison to adjacent normal tissue and not tumor-specific. Given the shared aspect of W>F substitutants, further investigations are still of interest [135].

#### 3.3.3. Termination of mRNA Translation

The highly efficient nature of translation stop codons has been known for over 25 years [129]. Thus, it is not surprising that, according to a review, noncanonical termination due to stop codon readthrough or ribosomal frameshifting at the termination codon is rarely observed [120]. In principle, it can lead to translation of aORFs containing the 3′ UTR, which might result in the formation of antigens. Indeed, another review proposed that such antigens could be of relevance for the generation of T-cell epitopes [122]. However, better understanding of noncanonical termination of mRNA translation in the context of cancer is required to determine the contained potential for immunotherapeutic approaches.

All in all, more research on the topic of noncanonical translation will enlighten future perspectives of respective treatment options. In particular, one review emphasizes that in vivo studies are necessary to elucidate if aberrantly translated neo-epitopes are expressed stably enough to generate a potent immune reaction that is capable of destroying the cancer [10]. Moreover, another review points out that the development of suitable prediction tools is needed to make clinical translation feasible [42].

### 3.4. Post-Translational Modifications

The final level at which neo-epitopes may be generated is post-translationally. For the majority of proteins, post-translational modifications (PTMs) play a fundamental role as PTMs can affect protein confirmation, stability, and localization. Since all these aspects have a crucial influence on protein function, dysregulation of PTMs is a frequent observation in the setting of cancer as stated in various reviews [137,138,139]. To date, around 300 PTMs have been discovered [140]. Here, the focus is set on those PTMs that are most relevant for the creation of neo-antigens in cancer (Figure 2A).

#### 3.4.1. Phosphorylation

The most disease-abundant and probably the most studied PTM is the phosphorylation of proteins [141]. When phosphorylated proteins (phosphoproteins) are degraded, the phosphorylation can remain on the arising peptides, generating phosphopeptides. Interestingly, it has been shown that phosphopeptides can harbor different antigenic features and a higher MHC-binding affinity compared to their cognate counter peptide. Both can favor the creation of neo-epitopes [142]. Further, complex formation with MHC has been found to protect phosphopeptides from dephosphorylation. This ensures stability of the phosphopeptides, which makes them particularly attractive as epitopes for T-cell-mediated therapies [143,144]. The ability of CTLs to recognize phosphopeptides has been demonstrated in human cell lines in vitro and in mice in vivo. It should also be noted that CTLs are capable of discriminating peptides from phosphopeptides [144]. A study comparing MHC-bound phosphopeptides in RCC and adjacent renal tissue identified one phosphopeptide which was detected in the RCC sample only. Thus, this phosphopeptide fulfilled the fundamental requirements of a neo-epitope, MHC-binding potential and tumor-specificity [145]. Further, neo-antigens were uncovered in a publication by Zarling et al. who focused on HLA-A2-associated phosphopeptides [146]. Here, only one of multiple discovered neo-antigens, pIRS2, shall be discussed. IRS2 stands for the insulin receptor substrate 2 (IRS2) and pIRS2 is formed through phosphorylation of the serine in position 1100. Hereafter, by (p)IRS2 we mean to refer to a (p)IRS2-derived (phospho)peptide. pIRS2 was present at high copy numbers in all three analyzed murine cancer cell lines but was absent in the noncancerous cell line. It is noteworthy that phosphorylation was crucial for the characterization as a neo-epitope. On the one hand, pIRS2 had four-fold increased MHC-binding affinity compared to the nonphosphorylated IRS2. On the other hand, only pIRS2 was expressed tumor-specifically, while IRS2 was also detected in the noncancerous cell line. It should be mentioned that the detection assay used for the latter point did not depend on (p)IRS2 binding to MHC. In addition to these neo-epitope features, pIRS2 possessed potent immunogenicity as demonstrated with a murine in vivo experiment. Within the scope of that, isolated CTLs showed a clear IFNγ response to pIRS2 but no response to IRS2, which underlines the pIRS2-specificity of the immune reaction. It is important to point out that the neo-epitope pIRS2 harbors the potential for the development of “off-the-shelf” treatment options, since it was detected in all cancer cell lines studied, that is, two melanoma and one ovarian carcinoma cell line [146]. In fact, the exact same neo-epitope has already been translated into a clinical trial in patients with high-risk melanoma (ClinicalTrials.gov number: NCT01846143). Alongside, a second phosphopeptide-neo-epitope, pBCAR3 (phosphorylated breast cancer antiestrogen resistance 3), was tested both separately and in combination with pIRS2. For both mono-treatments, three participants were enrolled while nine patients received the combined vaccine. Thus, twelve participants were treated with pIRS2 and pBCAR3 each. In total, the analysis of T-cell responses revealed five and two, out of twelve patients, responsive to pIRS2 and pBCAR3, respectively. In terms of safety, adverse effects of grades 1 or 2 were present in all patients, but no grade 3–4 adverse effects were seen whatsoever [147]. In summary, even though the strength of the induced immune response could be improved, the study indicates the potential of phosphopeptides in the context of neo-epitope-targeted vaccines. Moreover, it provides a promising perspective for the future of “off-the-shelf” vaccines.

#### 3.4.2. Glycosylation

Another PTM with implications for the formation of neo-antigens is the glycosylation of proteins. Glycosylation is known to influence tumor progression and dissemination [148,149]. One relevant group of highly glycosylated proteins are mucins since changes in their glycosylation pattern are associated with epithelial tissue cancers [150]. For example, breast cancer frequently harbors aberrantly glycosylated mucin 1 (MUC1) at upregulated expression levels [151]. In fact, a large multicenter phase III trial of a therapeutic MUC1-directed vaccine in metastatic breast cancer patients had already been started in 1998 (ClinicalTrials.gov number: NCT00003638). The target of the so-called Theratope vaccine was the carbohydrate antigen Sialyl-Tn (STn), which can be found on cancer-associated MUC1 [152,153]. Generally, STn is considered to be expressed tumor-dominantly [154,155]. Although the trial’s primary objectives in disease progression and overall survival were not met, the vaccine was well tolerated and led to specific anti-STn immunoglobulin G (IgG) responses [153]. Indeed, a correlation between IgG levels and median survival was present. Patients who exhibited an IgG titer above the median survived significantly longer than those with IgG levels below the median [153,156]. Unfortunately, the cellular immune reaction was not analyzed, even though a previous study had suggested the presence of STn antigen-specific T cells in Theratope-treated patients [157]. In spite of its shortcomings, this phase III trial was still described as the most promising approach of a glycan-targeted vaccine in a review in 2020 [158]. A different study investigating glycopeptides focused on a peptide derived from GalNAc-glycosylated MUC1 (GalNAc: N-acetylgalactosamine). Remarkably, the glycosylation increased the affinity of the peptide towards MHC by a factor of 100. Also, immunogenicity of the peptide was proven in murine in vitro and in in vivo experiments. However, the CTLs were not capable of differentiating between the glycosylated and the cognate peptides but were fully cross-reactive [159]. Other mucins which have been studied in the context of tumor antigens include MUC4 and MUC16. Both are suggested as a potential source of neo-epitopes for the treatment of pancreatic cancer [7,160].

Besides the named GalNAc glycosylation, another form of glycosylation worth elaborating on is GlcNAc (N-acetylglucosamine), since GlcNAc can play a role in tumorigenesis and tumor progression according to a review [161]. Promising work regarding glycopeptides as neo-epitopes has been accomplished by Malaker et al. [162]. The analysis of five leukemia patients (one AML, one ALL, and three CLL) revealed 33 GlcNAc-modified peptides, which were associated with MHC and absent in healthy controls (spleen, tonsil, and a noncancerous cell line). Noteworthy, seven of these neo-epitope candidates were shared between all five patients indicating potential for “off-the-shelf” immunotherapeutics. In vitro experiments with healthy donor PBMCs (peripheral blood monocytes) demonstrated that five out of seven peptides elicited a cytotoxic memory T-cell response. This suggests an immunosurveillance mechanism which prevents the development of leukemia in healthy subjects. Moreover, it is evidence for immunogenic neo-epitopes [162], which provides an encouraging perspective for the treatment of leukemia.

#### 3.4.3. Other Post-Translational Modifications

Interestingly, the most immunogenic neo-epitope found by Malaker et al. [162] (see last paragraph under Section 3.4.2) did not only carry a GlcNAc modification but also a methylation. The coincubation of T cells with autologous transformed B-cell lines revealed that T cells were capable of killing B cells which presented one of three different variants of this peptide: (1) the glycosylated peptide, (2) the methylated peptide, and (3) the combined glycosylated and methylated peptide. B cells presenting the unmodified peptide, however, were not recognized by the T cells. Thus, the PTM was fundamental for T-cell recognition; plus, T cells were able to discriminate between the post-translationally modified and the cognate antigen [162]. The latter encourages the further development of respective immunotherapeutics as it may limit the risk of autoimmune adverse effects.

Other post-translational modifications which have been mentioned in the context of cancer neo-epitopes include deamidation [163], citrullination, and ubiquitination [120]. Given its central role in the processing of antigens, the last may influence the production of neo-epitopes indirectly as reported in a review article [120]. Finally, the complex mechanisms of protein degradation and antigen processing can be affected in cancer cells in various ways, leading to a complete shift in the HLA-ligandome. The resulting TEIPPs, T-cell epitopes with impaired peptide processing, are a complex and extensive topic on which further information can be found elsewhere [164,165,166,167].

The treatment options related to the above-described mechanisms of epitope formation are summarized in Table 2.

## 4. Viral Neo-Epitopes

Next to dysregulated or mutated endogenous tumorigenic drivers, some malignancies are triggered by viral activators. The concept that viral infection can cause cancer was developed more than half a century ago and yielded the Nobel Prize for Harald zur Hausen for his work on the role of human papillomavirus in cervical cancer [168]. The viral antigens involved in malignant transformation represent another class of shared neo-antigens. They may appear as ideal antigens for tumor immunotherapy, as they are not found in the germline genome and are thus not subject to central tolerance. They are highly similar across different patients and, in contrast to point mutations, one antigen can give rise to several different epitopes. Their functional involvement in malignant transformation makes the development of antigen-loss variants improbable. A variety of viral antigens have been discovered, and many of them have been suggested or used in immunotherapeutic approaches. On the contrary, viruses have developed their own mechanisms to escape immune surveillance; thus, the immunogenicity of their antigens often turned out to be not as high as originally expected. Nevertheless, viruses, or parts thereof, that are present in tumor cells can cause the expression of neo-antigens, which derive from host DNA sequences that are not part of the canonical human transcriptome, for example, by integration of viral promotors in otherwise transcriptionally silent regions.

### 4.1. Human Papillomavirus

The human papillomavirus (HPV) is a textbook example for viral oncogenesis. It is associated with several tumors including anal cancer, cervical cancer, vaginal and vulvar cancer, penile cancer, and head and neck squamous-cell carcinoma (HNSCC). Different strains of HPV harbor different carcinogenic potential: HPV6 and 11 are considered low risk, whereas HPV16, 18, 31, 33, 35, 39, 45, 51, 52, 56, 58, and 59 are considered high-risk strains reviewed in [169,170]. HPV16 is mostly responsible for 50% of cervical cancers and even higher rates at other sites of tumorigenesis [171]. In general, HPV carries the oncogenes *E6*, that degrades p53, and *E7*, that disrupts pRb, summarized in [172]. The expression of the other ORFs, E1, E2, E4, E5, L1, and L2, support the viral antigens E6 and E7 in oncogenesis [173,174].

By comparing HPV-positive HNSCC samples with negative ones, it was shown that E6 and E7 are not sufficient for a malignant transformation. Nevertheless, an integration of the HPV genome can cause an E6-mediated degradation of TP53 causing genomic instability via rearrangements, translocations, amplifications, and ploidy in mouse models [175,176,177]. Furthermore, the usually antiviral activity of APOBEC drives mutations in single-strand DNA during genome replication in tumor cells and higher frequencies of deletions involving microhomology-based repair. This leads to an increased genomic instability that supports cancer development, as reviewed in [175].

For the prevention of HPV infection, there are two prophylactic vaccines available: Gardasil9^®^ and Cervarix^®^, which give hope for a decrease in numbers of HPV-induced malignancies in the future. Both vaccines are virus-like particles (VLPs) that induce strong cellular and humoral neutralizing immune responses against the major capsid protein L1 of HPV [169,178]. L1 is not conserved across different HPV strains and therefore the available vaccines only protect against certain strains and do not guarantee complete HPV protection. Although the minor capsid protein L2 could be an alternative target because it is found in all HPV strains, the antigen is less immunogenic than L1 and does not form VLPs on its own. Researchers are working on designing VLPs that display L2 peptides in order to broaden to range of prophylactic HPV vaccines, as reviewed in [179]. Prophylactic vaccines might reduce the chances of developing an HPV-induced disease, but they offer no clinical benefit for people that are already infected with HPV or even bear HPV-induced tumors. Smalley Rumfield et al. provide an extensive summary of all the therapeutic strategies used for vaccines to treat HPV-associated malignancies [169]. Treatments based on peptides, proteins, viral and bacterial vectors, DNA, RNA, and cells have been explored and some have already been tested in clinical trials.

Therapeutic vaccines based on short peptides bind directly to the HLA molecules on the surface of the cells of the patient, in most cases to HLA class I, and elicit CD8^+^ T-cell responses. An example is the PDS0101 vaccine based on six HLA-unrestricted peptides from HPV16 E6 and E7, which is administered with R-DOTAP, a TLR7-activating adjuvant. This safe vaccination is tested in combination with the immune checkpoint inhibitor Pembrolizumab in HPV16^+^ HNSCC (NCT04260126; active but not recruiting) or combined with the necrosis-targeted IL-12 immunocytokine (NHS-IL12) and bintrafusp (NCT04287868; active, but not recruiting) after successful results in mice [180,181]. The PDS0101/NHS-IL12/bintrafusp study showed an overall response rate of 22% (11/50) which increased up to 63% (5/8) in ICI-naïve patients with HPV16^+^ cancers. Treatment-related adverse events up to grade 4 were reported, but overall survival and progression-free survival were not specified in the results (published on clinicaltrials.gov [182]). Another clinical trial used a vaccine named CIGB-228, which is based on the HPV16 E7_86–93_ peptide adjuvanted with very small proteoliposomes (VSSP). Since CIGB-228 uses a known epitope for CTLs, the vaccination was able to induce HPV-specific memory T cells, which contributed to the high rate of complete regressions, 57.1%, in patients with cervical intraepithelial neoplasia [183].

Other therapeutic vaccines have combined short HPV peptides with non-HPV peptides like the melanoma antigen E (MAGE-A3) or human immunodeficiency viruses (HIV) peptides. Studies with HPV16 or MAGE-A3 peptides linked to HIV-TAT showed the safety, immunogenicity, and nontoxicity of the vaccination in HNSCC patients [184,185]. Another vaccine is based on DPX-E7 (fusion peptide of the HPV E7 peptide R9F_49–57_ and the universal helper epitope PADRE [186]), which is processed intracellularly to a small peptide. Here, the combination of the HPV16 E7 peptide with the DepoVax adjuvant showed decreased numbers of regulatory T cells and myeloid-derived suppressor cells in mice [187] and is now tested in a clinical trial (NCT02865135). Twenty-two days after vaccination, one patient was classified as a responder as measured with the increase in CD8^+^ T cells in peripheral blood and tumor tissue, but only 11 patients were enrolled in the study in total. There are also therapeutic vaccinations that are not based on the classical HPV E6 or E7 peptides. Due to the upregulation of p16 in HPV-associated cancers, the peptide p16_37–63_, together with the adjuvant Montanide^TM^, was used as a vaccination in two clinical trials (NCT01462838 and NCT02526316) [169,188].

Therapeutic vaccines that are based on long peptides need to be processed intracellularly before presentation and are more efficient in inducing CD4^+^ T-cell responses. An example for such a vaccine is ISA101, which consists of a total of 13 HPV16 E6 and E7 long synthetic peptides. In trials with severe cases of cervical cancer, vulvar intraepithelial neoplasia, and HPV16-associated gynecological carcinomas, the vaccine was well tolerated and elicited strong T-cell responses [189,190,191]. PepCan is a vaccine using a combination of four long HPV peptides and the adjuvant Candin^®^ (Allermed, San Diego, CA, USA) to induce HPV16 E6 specific immune responses and to decrease the viral load in patients with cervical intraepithelial neoplasia [192,193]. The treatment with the HPV peptides caused Langerhans cells to mature, whereas Candin^®^ induced an increase in T-cell proliferation [194]. PepCan is now being investigated in further trials with HNSCC patients (NCT03821272) and for the treatment of cervical squamous intraepithelial lesions (NCT02481414). The rate of complete and partial responses were lower in the PepCan/Candin^®^-treated patients with cervical squamous intraepithelial lesions (30.8% and 38.5%) compared to the cohort treated only with Candin^®^ (47.6% and 57.1%). The frequency of serious adverse events was comparable in both groups with 8% in the PepCan and 5% in the Candin^®^ cohort, whereas more than 94% of all patients in the trial experienced nonserious adverse events [182].

A third example for a therapeutic vaccine with a synthetic long peptide is called Hespecta (ISA201), where two HPV16 E6 peptides are administered with the TLR2 ligand Amplivant^®^. The combination of the peptides and the adjuvant induced strong antitumor immunity including dendritic cell (DC) maturation and in vivo T-cell priming in preclinical experiments in mice [195]. A clinical trial using Hespecta was started in 2017 and showed that the combination of TLR2-ligand and synthetic long peptides is a safe therapy that harbors potential to induce potent peptide-specific T-cell immune responses with acceptable side effects (NCT02821494) [182,196]. Other strategies combined synthetic long peptides with other adjuvants like CpG or nanoparticles to enhance the vaccination efficacy in preclinical murine models [197,198].

In HPV-induced HNSCC, the expression of PD-L1 on the tumor has been correlated with a decrease in overall survival of the patients, and, therefore, it seemed reasonable to combine therapeutic vaccinations based on HPV E6 and E7 peptides with immune checkpoint inhibitors. So far, there are multiple clinical trials already investigating this combination: NCT03439085 (active, not recruiting), NCT03946358 (active, not recruiting), NCT03618953 (terminated due to misjudgment of virus potency), NCT04001413 (terminated due to funding withdrawal), NCT04084951 (completed, no results yet), NCT04260126 (active, not recruiting), NCT02291055 (terminated, results available), NCT03260023 (recruiting), NCT03669718 (active, not recruiting), and NCT04369937 (recruiting) [169].

Cell-based vaccines used for the treatment of HPV-associated malignancies are all based on the HPV E6 or E7 proteins. Clinical trials with peptide-pulsed peripheral blood mononuclear cells (PBMCs) or DCs are conducted in patients with different HPV^+^ cancers, but results are yet to be published (NCT00003977, NCT00019110, NCT00155766, and NCT03870113), as summarized in [169]. Adoptive cell therapies in HPV^+^ HNSCC are used either via infusion of enriched HPV E6- and E7-specific TILs or via infusion of T cells expressing a T-cell receptor that recognizes HPV E6 or E7 peptides, as reviewed in [199]. Studies for both approaches have shown good clinical efficacy [200,201,202,203]. A disadvantage of these therapeutic vaccinations is the restriction that the peptides are only presented on HLA-A02:01. The discovery of two HPV E2 epitopes presented on HLA-A01:01 is therefore a chance to broaden the applicability of the adoptive cell transfer strategies [204].

### 4.2. Human T-Cell Leukemia Virus

The human T-cell leukemia virus type 1 (HTLV-1) is a retrovirus that has infected between 5 and 10 million people worldwide, but most infections are asymptotic. Nevertheless, 3–5% of infected people are developing adult T-cell leukemia or lymphoma (ATLL), as reviewed in [205,206], with the epidemiology of HTLV-1 and ATLL overlapping in southwestern Japan, parts of the Caribbean, South America, intertropical Africa, and other small isolated clusters [206]. According to Shimoyana, there are four classes of ATLL, all with different clinical features: the acute ATLL and the lymphoma ATLL are aggressive subtypes and make up the majority of all ATLL cases, whereas the chronic and smoldering subtypes are less frequent and less severe in their disease progress [206,207].

HTLV-1 integrates specifically at a nonpalindromic DNA motif [208], and during the latency state, viral proteins are only expressed via mitotic and clonally expanding infected cells. The virus not only is detectable in the blood in T cells that are CD4^+^ CD25^+^ CCR4^+^ CADM1^+^, but also in hematopoietic stem cells that act as a latent reservoir for the virus, as summarized in [209,210,211]. The immunodominant viral Tax protein, a transactivator protein for viral gene expression, is usually not found in PBMCs, but cytotoxic T cells (CTLs) in latently infected people are able to respond to Tax, although their responses are weak [209]. CD8^+^ T cells that are specific to the antisense protein HTLV-1 basic leucine zipper (HBZ) showed more effective responses, but it is suspected that the antigen presentation of HBZ is impaired by nuclear retention [209].

The treatment options for patients with relapsed or primary refractory ATLL are limited, and, when available, the participation in clinical trials, for example with therapeutic vaccines, is recommended [206]. A Japanese pilot study with Tax peptide-pulsed DCs seemed very promising (UMIN000011423) [212]. Synthetic oligopeptides (Tax_11–19_ LLFGYPVYV and Tax_301–309_ SFHSLHLLY) were created, based on the major epitopes of Tax-specific CTLs of ATLL patients post allo-hematopoietic stem cell transplantation (allo-HSCT) [213,214,215]. The vaccine was deemed safe and showed very promising results with improvements for all patients, and even a full remission was observed. According to Kannagi et al., the Tax-pulsed DC vaccine is currently under phase I trial [216]. Another Tax-directed T-cell immunotherapy is focusing on supporting CTLs in their function. Kawamura et al. demonstrated in mice that PBMCs that were transduced with a T-cell receptor specific to Tax_301–309_ by a retroviral siTCR vector, produced sufficient cytokines, and killed HTLV-1 infected T cells. This harbors the potential for a new immunotherapy for ATLL [217]. TheraVectys has developed an ATLL vaccine called THV02 that is based on two lentiviral vectors that are used to prime and boost HTLV-1-specific immune responses. They used a polypeptide that was derived from the viral proteins Tax, HBZ, p121, and p30II to induce HTLV-1 specific CD8^+^ T cells. After preclinical evaluations, the vaccine was shown to be safe with a limited diffusion, a fast clearance, no dissemination after injection, and efficient induction of T-cell responses. A clinical trial was planned in 2015 with patients in France, the UK, French Guiana, Martinique, and Guadeloupe, but no updates have been published since then [218].

While these three vaccines are based on HLA-specific peptides, a different vaccination approach with short-term cultivated autologous PBMCs aims to avoid this HLA restriction. Here, the ex vivo cultivation of PBMCs of the ATLL patient triggered the expression of Tax due to the missing suppressive stimuli. When cocultured with antigen-presenting cells, these PBMCs could serve as an antigen source for cross presentation of the Tax-antigen, thus provoking Tax-specific CTLs [219]. Another approach that avoids the HLA restriction of peptide-based therapies is a vaccination with a live attenuated varicella-zoster (VZV) vaccine. As occasionally observed, the development of a herpes zoster resulted in an increase in Tax-specific CD8^+^ T-cell levels in ATLL patients. To translate this observation to a clinical application, a commonly available varicella vaccine was utilized. This VZV virus activated HTLV-1 Tax-specific CTLs, which supported the efficacy of standard treatments for three patients with aggressive forms of ATLL [220].

### 4.3. Epstein–Barr Virus

The Epstein–Barr virus (EBV) is involved in malignant transformation in several tumors including Burkitt’s lymphoma (BL), nasopharyngeal carcinoma (NPC), and Hodgkin’s lymphoma (HL). Mainly persisting in B cells in a latency state, EBV tumorigenesis generally only depends on the expression of the nuclear antigens EBNA1, 2, 3A, 3B, 3C, and –LP and the latent membrane proteins LMP1, 2A, and 2B, as summarized in [221]. As a trigger of latency into the lytic cycle, the immediate early proteins BZLF1 and BRLF1 have been established.

EBV-associated cancers have a specific expression pattern of different viral genes and can therefore be classified in three categories, as reviewed in [221]. The least immunogenic category is the type 1 latency cancer that includes BL. Here, only the viral EBNA1 is expressed. NPC, gastric carcinoma, HL, and natural killer cell lymphoma are grouped into the type 2 latency cancers with their characteristic expression of EBNA1, LMP1, LMP2, and BARF1, which are intermediately immunogenic. Type 3 latency cancers include post-transplant lymphoproliferative disorder of hematopoietic stem cell transplants, solid organ transplant recipients, and B-cell lymphoma in AIDS patients, and they express the viral proteins EBNA1, EBNA2, EBNA3A, EBNA3B, EBNA3C, LP, LMP1, LMP2, and BARF1 as well as noncoding RNAs, miRNAs, and small RNAs. Here, the high immunogenicity of the tumors arises from the expression of the EBNA3 proteins [222,223]. However, since these tumors occur under immunocompromised conditions, no effective immune response arises.

There are already several trials targeting latent EBV antigens, as reviewed in [221,224]. Adoptive T-cell transfer approaches based on the immunogenicity of EBNA3 have proved to be favorable in several trials in post-transplant lymphoproliferative disorder (PTLD) but could not overcome the immunosuppressive environment of the tumor [225]. Here, genetic modifications are thought to improve the efficacy of the treatment. Furthermore, T cells specific for EBNA1, LMP1, and LMP2 showed promising results in the less immunogenic type 2 latency tumors NPC and HL [226,227,228,229,230]. By broadening the T-cell specificity to BARF1, the treatment could become even more effective [231]. TCRs recognizing the antigens EBNA3A, EBNA3B, LMP1, LMP2, BRLF1, and BMLF1 were cloned, and functional responses against EBNA3A, LMP1, LMP2, and BRLF1 were shown in vitro [232,233,234,235]. LMP1-specific TCR-engineered T cells delayed tumor growth in a murine xenograft leukemia model [232]. Although only a weak recognition of EBV-transformed cell lines by T cell expressing a LMP2-specific TCR was found in vitro, significant resistance to LMP2-positive tumor cell challenge in a NPC nude mouse model was detected after injection of these T cells [235]. The efficacy could be improved by stabilizing the TCR by a chimeric design, and TCR engineered T-cell therapy remains a promising strategy [221]. An example for this would be the introduction of a CD28 domain in the EBV-specific TCR to enhance the interferon-gamma production by the antigen-specific T cells [236]. Besides adoptive T-cell therapy, there have also been several trials using EBV antigens in DC-based vaccination approaches for EBV-associated cancers. Multiple studies have used DCs pulsed with EBV peptides from LMP2, LMP2A, and LMP1 in NPC patients, but the efficacy and persistence needed to improve [237]. Based on these studies, a phase I study was conducted with CD137L-expressing DCs that were pulsed with a peptide mix of EBV LMP1, LMP2, and LMP2A as a vaccination for NPC patients. Here, patients with a recorded clinical benefit (42%, 5/12) showed an increased ratio of CD8^+^ memory T cells to naïve T cells compared to nonresponders. It is noteworthy that high numbers of CD8^+^ CD4^+^ T cells prior to the first vaccination correlate with the responder cohort, which could be used as prognostic marker for NPC disease progression in the future (NCT032826179).

In the context of BL, the ideas of utilizing EBV antigens for immunotherapy has to be taken with a grain of salt. Although the world health organization (WHO) has declared EBV as the causative agent of BL due to the high antibody titers to viral antigens of the lytic EBV replication cycle (viral capsid antigen and early antigen) in patients [238], the low or nonexistent expression of immunogenic viral antigens in BL decreases the application of viral-based immunotherapy for BL.

### 4.4. Cytomegalovirus

In human colorectal cancer, EBV-negative HL, intraepithelial neoplasia, prostatic carcinoma, and high- and low-grade gliomas, cytomegalovirus (CMV) DNA or protein expression was detected, although none of these cancers is known to be CMV-induced [239]. Especially in glioblastomas, CMV signatures were found exclusively in the tumor tissue compared to the surrounding brain tissue and therefore offer a unique targeting strategy.

Different research groups have found varying levels of CMV proteins in the blood of glioblastoma patients and in the tumor itself, which was attributed to the sensitivity of the used assays, as reviewed in [239,240]. Although the role of CMV in carcinogenesis is not yet understood, there is a consensus that most human gliomas express CMV antigens, as summarized in [240]. Examples for CMV proteins expressed in glioblastoma are pp65 and IE1 [241,242]. Although IE1 expression correlates with an infiltration of T cells, these antigen-specific T cells expressed the immunosuppressive checkpoints CTLA-4 and PD-1 and responded with decreased IFNγ production upon peptide stimulation. This indicates that the immunogenic CMV antigens can induce antigen-specific T cells but those are tolerized [243].

There have been several therapeutic approaches to glioblastoma treatment by targeting CMV, including adoptive T-cell transfer, DC vaccines, and VLP vaccines. CMV-specific T cells were isolated from glioblastoma patients and expanded ex vivo. The specificity of the CMV antigens of each patient was determined, and the isolated T cells were stimulated with the patient-specific CMV peptides. Adoptive T-cell transfer back into the patient demonstrated a good outcome, as reviewed in [239]. Other studies have tested the efficacy of DC vaccines that were loaded with CMV *pp65* RNA (NCT02465268). In a mouse model, an adenoviral vector that specifically induces the expression of CMV-IE in DCs resulted in a survival benefit [244]. The combination of DC vaccines with either adoptive T-cell transfer or dose-modified chemotherapy appears promising as an alternative treatment strategy for glioblastoma, as summarized in [239].

In early phases of glioblastoma disease progression, the CMV genome and protein expression is relatively low, and, therefore, the high efficiency of CMV-based immunotherapy against glioblastoma has been surprising, and three hypothetical explanations were brought forth. The direct antigen-specific hypothesis proposes that the high efficacy is due to the stem cell-like phenotype of the CMV-expressing tumor cells and assumes that by eliminating the CMV-infected cells, the cells responsible for growth and resistance establishment are specifically targeted, which inhibits the tumor growth. The indirect targeting hypothesis supports the idea that by killing the CMV-expressing tumor cells, other immune cells become activated and secrete antitumor cytokines to fight the tumor cells further. The third hypothesis is based on cross-priming where the release of antigens of killed CMV-tumor cells triggers the priming of cytotoxic T cells by DCs that have taken up these antigens. A combination of all of these three proposed mechanisms is possible and most likely happening in CMV-based immunotherapy against glioblastoma, as reviewed in [239].

### 4.5. Merkel Cell Polyoma Virus

The Merkel-Cell Polyomavirus induces around 80% of Merkel-cell carcinoma (MCC) incidents which display an overall low tumor mutational burden compared to nonvirus-induced MCC, as reviewed in [245]. For a malignant transformation, the viral genome integrates at random sites into the host genome, for example, via viral fragmentation during viral replication. Furthermore, the viral genome needs to acquire a specific mutation, which causes a truncation of the viral Large T (LT) antigen. This mutation can be caused by UV-light exposure, and it has been suggested that the mutation is acquired before or during the integration of the viral genome, but to this point, neither the chronological order or the mechanism of the specifics of the mutation origin or the integration process has been shown, as summarized in [245,246,247].

Although treatment with immune checkpoint inhibitors like Avelumab (anti-PD-L1) or Pembrolizumab (anti-PD-1) has improved the treatment of MCC, almost 50% of patients never respond to ICI or evolve a resistance against the immunotherapy, as reviewed in [245]. Therefore, alternative strategies are needed, and a vaccination based on MCPyV-specific neo-epitopes could be an option. Already, in 2011, several epitopes based on the MCPyV capsid proteins and the oncoproteins were identified from MCC tumors as well as from MCC patient blood [248]. Later, Lyngaa et al. found CD8^+^ T cells responding to 35 different MCPyV-peptide sequences only in the MCC patient cohort compared to healthy controls [249]. Preclinical evaluations of a vaccine based on DC transfected with truncated LT mRNA showed an induction of antigen-specific CD8^+^ T cells from blood of healthy donors and even higher numbers with cells derived from MCC patients, and it holds potential for adoptive cellular immunotherapy [250]. Others have analyzed the TILs in MCC and could identify several additional virus-specific epitopes [251]. Recently, there have been another eleven new epitopes found that are derived from the T antigens and support the recognition of MCC tumor cells by antigen-specific T cells [252]. These findings suggest that a vaccination based on MCPyV-specific neo-epitopes could become a new treatment strategy for MCC, especially in a combination approach with ICI.

### 4.6. Kaposi’s Sarcoma Assoiated Herpesvirus

Kaposi’s sarcoma-associated herpesvirus (KSHV, HHV-8) is found in primary effusion lymphoma (PEL), Kaposi’s sarcoma (KS), and multicentric casteleman’s disease (MCD). The latency-associated nuclear antigen (LANA) is known as the universal marker for KSHV latency, and, therefore, an immunotherapy targeting LANA-expressing cells would be beneficial to treat KSHV-induced malignancies. Several isoforms of LANA generated via internal translation initiation, premature termination, internal frameshifting, or proteolytic cleavage have been reported, as summarized in [253,254,255,256]. Unfortunately, KSHV proteins including LANA generally only elicit weak T-cell responses. Nalwoga et al. showed in a cohort of 116 KSHV seropositive participants in Uganda that the interferon-gamma responses to KSHV peptides were of low intensity and infrequent compared to other herpesviruses, like EBV or CMV [257]. These results are similar to studies with KS patients and healthy donors [258,259,260]. Some individuals responded intensively to the envelope glycoprotein or LANA of KSHV, but the response did not correlate with factors like sex or age. Nevertheless, the observed responses to KSHV seemed sufficient to control the onset of any KSHV-related disease [257].

Research was able to show that the KSHV proteins play a role in the immune evasion process. For example, the viral ubiquitin ligases K5 and K3 and transcription factor RTA are involved in the degradation of MHC class I and class II molecules, CD1d, CD31, CD54, B7-2, MICA, MICB, and IFNγR1. This leads to a reduced functionality of CD4^+^ T cells and CD8^+^ T cells, as well as NK cells, which are all associated with antitumor immune responses, as reviewed in [261]. LANA also contributes to the escape of KSHV-infected cells from the immune system. The aspartate- and glutamate-rich elements in the internal repeat region of LANA decrease the loading of viral peptides on MHC class I molecules. LANA is also able to bind to IRF4 and RFX, which both reduce the presentation of peptides on MHC class II molecules. Furthermore, LANA influences the antiviral interferon and NFκB signaling pathways by inactivating innate immune sensors, as reviewed in [253]. Therefore, strategies for immunotherapies based on neo-antigens on tumor cells have been limited. So far, the KSHV antigens gB, K8.1, LANA, and K12 were loaded on DCs to stimulate CD8^+^ T cells to create a T-cell line that can be used for adoptive T-cell transfer. Although KSHV can directly infect DCs, vaccines based on such DCs are not a promising therapy strategy due to the decreased IL-12 production upon the loading with KSHV antigens, which reduces the T-cell-priming efficacy, as reviewed in [262].

### 4.7. Hepatitis B and C Virus

Hepatitis B virus (HBV) is the predominant reason for hepatocellular carcinoma (HCC) in Eastern Asia, South-Eastern Asia, and Africa. In contrast, in the USA, Europe, Japan, and South America, a chronic infection of Hepatitis C virus (HCV) is the main cause of HCC, as reviewed in [263,264,265]. Worldwide, in 2020, approximately 906,000 people were newly diagnosed with HCC and around 830,000 deaths of HCC patients were registered [266]. The pathogenesis of and different treatment strategies for HCC are reviewed in detail by Tümen et al. [264]. Here, we focus on possible neo-epitopes in the tumor tissue, specifically in HBV-induced HCC.

Apart from the normal HBV replication cycle, the viral DNA can integrate into the host genome by double-strand DNA breaks. This stops viral replication and results in deletions of up to 200 bp in the integrated viral genome. The sites of viral integration in the genome can influence endogenous genes like *TP53*, *TERT1*, and *CTNNB1* and major signaling pathways, for example, Wnt/ß-catenin and JAK/STAT, as well as changes in the epigenetic landscape of the human genome. Therefore, it is thought to be a major driver in malignant transformation [264]. Sequencing of HCC tissue revealed that the viral gene *HBX* frequently harbored nonsense and frameshift mutations, as well as deletions [267]. These mutations often led to a truncation and a subsequent loss of the stop codon, which could cause HBV cellular fusion transcripts like *HBx-LINE1* or *HBx-MLL4* [268,269]. Furthermore, many HBV-related HCC cells did not contain the whole HBV genome, but rather integrated fragments of the viral DNA that led to short viral mRNAs. These fragmented mRNAs encode epitopes that are recognizable by T cells [270], and recent studies have focused on identifying new HBV-derived epitopes [271,272].

The genomic alterations to the human genome by the integration of HBV DNA also can cause expression of neo-antigens due to expression of genomic sequences, which are not translated in healthy cells. Here, a viral promoter integrates into a noncoding region of the human genome resulting in the transcription and subsequent translation of peptides from the noncoding region. An example for HBV is the integration of the viral genome in the intron region of the *CCNA2* gene. Since viral integration acts as an oncogenic driver for HCC and expression of viral peptides remains stable across the disease progression in HCC, an immunotherapy targeting those viral neo-antigens could be a possible treatment strategy, as reviewed in [21].

Several vaccination treatment strategies against neo-antigens in HCC have been explored but most trials had limited success, as reviewed in [21,273]. A promising outlook is given by the study with the HepaVac-101 vaccine, previously in trials for renal cancer, which evaluated the safety and immunogenicity of 17 tumor-associated peptides combined with the TLR7/8/RIG I agonist CV8102 in patients with HCC (NCT03203005). The peptides were identified via mass spectrometry in human HCC tissue or cell line, and the vaccine included five peptides that were specific to HLA-A24, seven to HLA-A02, and four to HLA-DR. The phase I/II trial showed a good safety profile with moderate induction of peptide-specific immune responses [274]. Further clinical trials are planned.

Nevertheless, the presence of neo-antigens seems only to be beneficial when the TILs have a high cytotoxic functionality, which is often reduced in the abundant exhaustion T-cell state found in tumors. The expression of immune checkpoints like PD-1 and CTLA-4 has been found on these exhausted TILs, as summarized in [21]. A combination treatment of neo-antigen-based vaccination with immune checkpoint inhibitors could therefore also be a strategy to improve efficacy of HCC treatments. Research focuses now on identifying new HCC neo-antigens, and the analysis of new high-throughput sequencing brings hope for new neo-antigens [21,264].

### 4.8. Human Endogenous Retroviruses

Human endogenous retroviruses (hERVs) are usually not able to replicate but there has been evidence that expression can occur tissue-specifically as well as in malignant cells, as reviewed in [275]. Retroviral elements include HERVs, LINEs, or SINEs [276,277]. The human ERV1 is, for example, associated with renal cancer [278], and the HERV-K-MEL is highly expressed in HNSCC [279]. The presented retrovirus-specific antigens have been shown to elicit potent T-cell responses [78,280,281].

Cherkasova et al. detected HERV-E envelope peptides that were presented on clear-cell renal carcinoma cells (ccRCC) and were able to induce CD8^+^ T cells [280]. A different study predicted further HERV epitopes specific to ccRCC in silico, validated their binding capacity to HLA, and showed an increase in these HERV sequences in patients responsive to PD-1 inhibition compared to nonresponders [78]. Rycaj et al. investigated a DC-based vaccination approach in the context of cervical cancer where DCs were transfected with HERV-K ENV or HPV16 E6 cRNA. The vaccine was able to induce typical antitumor immune responses including antigen-specific CTL activity, which was more pronounced in cells from ovarian cancer patients compared to samples from patient with benign diseases [281]. A treatment strategy based on HERV elements might seem promising especially in combination with ICI, but the identification of potential target peptides is difficult due the highly repetitive genomic structure as well as a possible expression in healthy tissues.

The mechanisms of integration and the roles of the above-described viruses in tumorigenesis together with the respective antigens are summarized in Table 3.

## 5. Conclusions

Neo-epitopes are the tags by which the adaptive immune system can best differentiate malignant cells from healthy tissue. While most canonical neo-epitopes are unique to individual patients and thus their exploitation requires highly personalized treatment strategies, noncanonical neo-epitopes are more often generated by general mechanisms and are often shared between patients. This allows for a broader application circumventing the difficulties that come along with highly personalized therapies. However, a deeper understanding of the underlying mechanisms is needed to exploit the therapeutic potential of these neo-epitopes. Viral tumorigenic drivers are another source of neo-epitopes, which are mostly shared between individual patients. However, the complexity of tumorigenesis via different viruses and the various functions of viral proteins requires additional research to find suitable immunotherapeutic strategies.

## Figures and Tables

**Figure 1 ijms-25-04673-f001:**
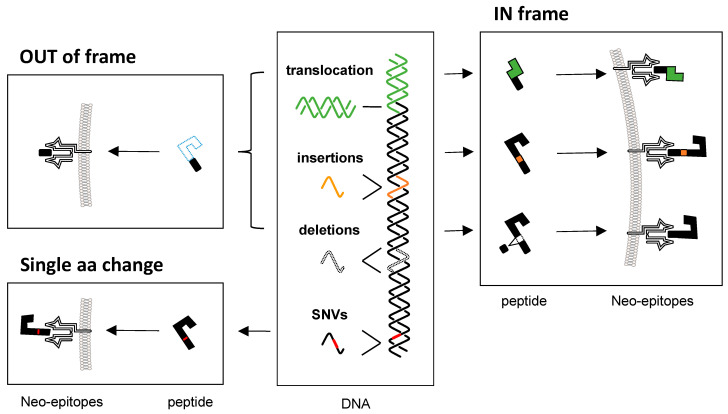
Illustration of different mechanisms in the formation of canonical neo-epitopes. Mutations include translocations, insertions, deletions, and single-nucleotide variants (SNVs). These first three types can alter the reading frame resulting in neo-epitopes which are partially or completely out of frame. In-frame translocations generate peptides, fused from two different genes, insertions generate peptides containing a stretch of additional amino acids and deletions fuse parts of proteins, which are normally separated. SNVs generate neo-epitopes in which a single amino acid is exchanged.

**Figure 2 ijms-25-04673-f002:**
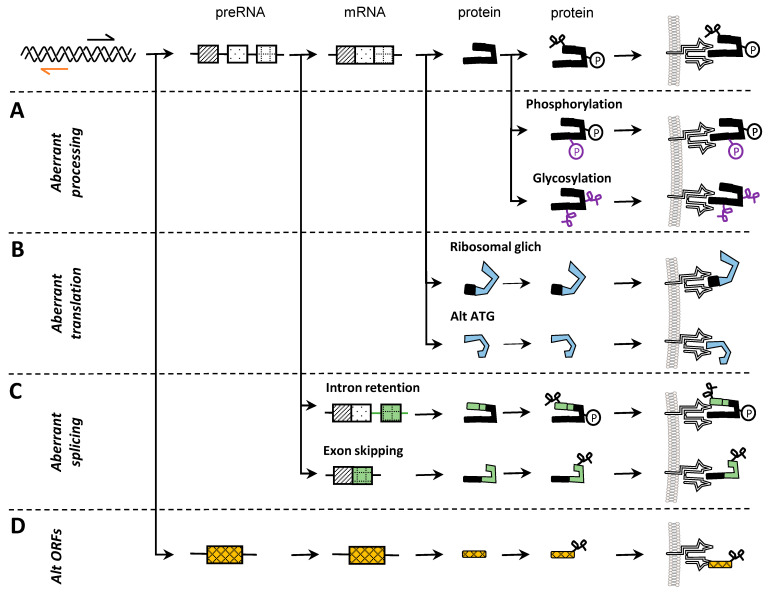
Illustration of different mechanisms in the formation of noncanonical neo-epitopes. The top depicts the formation of a peptide epitope that has not been altered. Noncanonical epitopes do not necessarily derive from mutation events. (**A**) Post-translational modification like altered glycosylation or phosphorylation may persist in processed peptides. (**B**) Aberrant translation may start from alternative ATGs (Alt ATG), and ribosomes may slip during translation (ribosomal glich), resulting in frameshifted amino acid sequences. (**C**) Aberrant splicing may lead to partially or completely different neo-epitopes via translation of intron sequences or exon skipping. (**D**) Alternative open reading frames (Alt ORFs) can be transcribed from cryptic promotors, inactive in normal cells.

**Table 1 ijms-25-04673-t001:** Based on the mechanism of the formation of neo-epitopes from the canonical pathway, the available treatment strategies in the relevant cancers are summarized.

Mechanism of Formation	Treatment Options Based on Neo-Antigens	Cancers	Reference
nsSNV *	Personalized treatment: passenger mutations, e.g., mRNA-4157/V940 vaccine	Cutaneous melanoma with LN metastases	[41]
	Off-the-shelf treatment: driver mutations, e.g., ACT HLA-C*08:02 KRASG12D-specific TCR data	Metastatic pancreatic cancer, Colorectal cancer with lung metastases	[47,49]
Indels	Personalized treatment	ccRCC and other	[26,57]
	Off-the-shelf treatment: MMR deficiency	MSI-H associated cancers: colorectal, endometrial, stomach cancer, etc.	[61,62,63]
Gene fusions	Personalized treatment	CML, synovial sarcoma	[72,73,74,75]

* Abbreviations: nsSNV = nonsynonymous single-nucleotide variant, LN = lymph node, ACT = adoptive cell therapy, HLA = human leukocyte antigen, TCR = T-cell receptor, CML = chronic myeloid leukemia, ccRCC = clear-cell renal-cell carcinoma, MSI-H = microsatellite-instable high, MMR = mismatch repair.

**Table 2 ijms-25-04673-t002:** Based on the mechanism of the formation of neo-epitopes from the noncanonical pathway, the available treatment strategies in the relevant cancers are summarized.

Mechanism of Formation		Treatments Based on Neo-Antigens	Cancers	Reference
Noncoding DNA		Off-the-shelf treatment, e.g., hERVs *, pseudogenes, and lncRNA	ccRCC, NSCLC, breast cancer, and MMR-proficient colorectal cancer	[78,81,82,84]
Splicing	*Cis*-acting mutations	Personalized treatment: splice site mutations	NSCLC	[94]
	*Trans*-acting mutations	Off-the-shelf treatment: splice factor mutations, e.g., SF3B1 mutation	Breast cancer, ovarian serous carcinoma, and hematological malignancies	[91,113]
	Pharmacologic splicing inhibitors	Off-the-shelf treatment, e.g., Indisulam	Colon and lung cancer (mouse model)	[115]
Translation	Initiation	Off-the-shelf treatment: near-cognate start codon	RCC	[130]
	Elongation	Off-the-shelf treatment: sloppiness codon reassignment	Melanoma, skin, breast, ovarian, lung, colorectal cancer, hepatocellular, head and neck squamous-cell carcinoma, and lung cancer	[120,122,133,134]
	Termination	Rarely observed, clinical consequence unclear		[135]
PTMs	Phosphorylation	Off-the-shelf treatment: phosphopeptides	Melanoma and ovarian carcinoma	[146,147]
	Glycosylation	Off-the-shelf treatment: glycosylated mucins	Breast cancer and pancreatic cancer	[7,153]
		Off-the-shelf treatment: GlcNAc	AML, ALL, and CLL	[162]

* Abbreviations: hERV = human endogenous retrovirus, lncRNA = long noncoding RNA, ccRCC = clear-cell renal-cell carcinoma, NSCLC = nonsmall-cell lung cancer, MMR = mismatch repair, RCC = renal-cell carcinoma, PTM = post-translational modification, GlcNAc = N-acetylglucosamine, AML = acute myeloid leukemia, ALL = acute lymphocytic leukemia, CLL = chronic lymphocytic leukemia.

**Table 3 ijms-25-04673-t003:** The mechanisms of integration of different viruses and the role of the integrated virus for carcinogenesis are summarized. The resulting virus-specific neo-antigens are also depicted.

Virus	Mechanism of Integration/Role in Carcinogenesis	Viral Neo-Antigens
HPV *	Integration causes genomic instability such as rearrangements, translocations, amplifications, and ploidy changes	E6, E7
HTLV-1	Integration at nonpalindromic DNA motif; during latency state, only viral protein expression by mitotic and clonally expanding infected cells	Tax, HTLV-1 basic leucine zipper, p121, p30II
EBV	Switch from latency into lytic replication triggers malignant transformation	EBNA1, EBNA2, EBNA3A, EBNA3B, EBNA3C, EBNA-LP, LMP1, LMP2A, LMP2B, BZLF1, BRLF1
CMV	Role of CMV in carcinogenesis not yet understood	pp65, IE1
MCPyV	Integration at random sites, e.g., via viral fragmentation during replication; mutation in viral LT before or during viral integration	LT
KSHV	Isoforms via internal translation initiation, premature termination, internal frameshifting, or proteolytic cleavage	LANA, gB, K8.1, K12
HBV and HCV	Integrates via double-strand DNA breaks which causes mutations in the viral genome causing truncations or cellular fusion proteins Integration of viral genes into intron region of CCNA2 leads to transcription of noncoding region	HBX, CCNA2 (endogen)
hERV	Endogenous integration; role in carcinogenesis not yet understood	ERV1, HERV-K-MEL, HERV-E

* Abbreviations: HPV = human papillomavirus, HTLV-1 = human T-lymphotropic virus 1, EBV = Epstein–Barr virus, EBNA = Epstein–Barr nuclear antigen, LMP = latent membrane protein, CMV = cytomegalovirus, IE = immediate early protein, MCPyV = Merkel-cell polyomavirus, LT = large T antigen, KSHV = Kaposi’s sarcoma-associated herpesvirus, LANA = latency-associated nuclear antigen, HBV = Hepatitis B virus, HCV = Hepatitis C virus, hERV = human endogenous retrovirus.

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
