# Peer review of "Tumor Antigens beyond the Human Exome"

_ijms, 2024, doi:10.3390/ijms25094673_

Round 1
Reviewer 1 Report
Comments and Suggestions for Authors
The manuscript is so comprehensive and well-written; however, I strongly suggest inserting a table encompassing the neo-antigen, the mechanism of formation and clinical consequence of such modification/dysregulation.
Figure 2 is not well-drawn. It must be revised in a more informative manner and divide in separate parts. Each part must also be described in figure caption separately.
A distinct figure describing the mechanism of viral neo-epitope formation also seems necessary.
Reviewer 2 Report
Comments and Suggestions for Authors
I enjoyed this comprehensive and well written review.
I have some minor comments/suggestions:
1. Reference to reviews – it is important to note in the text when you are referencing a review (rather than original study), this is only fair to the reader and avoids the dreaded perpetuation of misquoting. I note the throughout the manuscript you sometimes correctly note that a reference is to a review, but mostly not.
2. Reference to animal studies – it is important to clarify in the text when the original work you site is done in animal models. Once again, I note that you sometimes do this in the manuscript, but mostly not. This is probably worse than not doing it all because it leads the reader to assume that when it is not mentioned the reference is to a human study.
3. Reference 159 is a review but the sentence on lines 728-730 reads as though it is the original study.
4. “Tax” first appears on line 915 but there is no explanation of what that is.
My final suggestion is that this excellent body of work would be much easier to digest and have wider appeal if it was accompanied by summary tables where possible.
Round 2
Reviewer 1 Report
Comments and Suggestions for Authors
Thanks for the rectification.